# The Problem of Monitoring the Psycho-Physical Condition of Seniors during Proposed Activities in Urban Space

**DOI:** 10.3390/s23031602

**Published:** 2023-02-01

**Authors:** Ewa Lach, Anna Szewczenko, Iwona Chuchnowska, Natalia Bursiewicz, Iwona Benek, Sylwia Widzisz-Pronobis, Daria Bal, Klaudia Elsner, Marta Sanigórska, Mateusz Sutor, Jakub Włodarz

**Affiliations:** 1Faculty of Automatic Control, Electronics and Computer Science, Silesian University of Technology, 44-100 Gliwice, Poland; 2Faculty of Architecture, Silesian University of Technology, 44-100 Gliwice, Poland; 3Faculty of Biomedical Engineering, Silesian University of Technology, 44-100 Gliwice, Poland; 4Institute of History and Archival Studies, Pedagogical University of KEN in Cracow, 30-084 Krakow, Poland; 5Joint Doctoral School, Silesian University of Technology, 44-100 Gliwice, Poland

**Keywords:** elderly, active ageing, monitoring, mobile application, wearable sensors, active urban space

## Abstract

The world’s population is rapidly ageing, which places a heavy burden on traditional healthcare systems with increased economic and social costs. Technology can assist in the implementation of strategies that enable active and independent ageing by promoting and motivating health-related behaviours, monitoring, and collecting data on daily life for assessment and for aiding in independent living. ICT (Information and Communication Technology) tools can help prevent cognitive and physical decline and social isolation, and enable elderly people to live independently. In this paper, we introduced a comprehensive tool for guiding seniors along the designed urban health paths employing urban architecture as an impulse to perform physical and cognitive exercises. The behaviour of seniors is monitored during their activities using wearable sensors and mobile application. We distinguished three types of data recipients (seniors, path/exercise designers, and the public), for whom we proposed methods of analysing the obtained data and examples of their use. In this work, a wide range of diverse information was examined from which short- and long-term patterns can be drawn. We have also shown that by fusing sensory data and data from mobile applications, we can give context to sensory data, thanks to which we can formulate more insightful assessments of seniors’ behaviour.

## 1. Introduction

The world’s population is rapidly ageing. In 2018, persons aged 65 years or over outnumbered children under five years of age worldwide for the first time. By 2050, one in every four persons in Europe and North America could be 65 or older. Populations in other regions are also projected to age significantly over the next several decades [1]. As we age, we see a decline in cognitive, physical, and social functions. Functional decline adversely affects the ability to independently perform the basic and more complex activities of daily living (ADL). There are many health detriments associated with an ageing population, such as strokes, diabetes, and falls. With the inversion in the demographic pyramid, there is also an accompanying increase in chronic diseases [2]. Currently, in European countries, assistance and support to the elderly are mostly provided by informal caregivers (e.g., relatives, friends) and formal caregivers (i.e., care professionals). This model places a heavy burden on traditional healthcare systems with increased economic and social costs. According to the Association of American Medical Colleges, demand for physicians continues to grow faster than supply, leading to a projected shortage, in the United States, of between 37,800 and 124,000 physicians by 2034 [3].

To address the challenges of an ageing population, both research and policy planning around the world are focused on ensuring a better quality of life for older people and their families and reducing the burden on the economy. This has started a trend in healthcare to go from curative to preventive care. In line with the concept of prevention being better than cure, more emphasis has been placed on preventing or delaying the onset or progression of diseases by promoting a healthy lifestyle [4,5,6,7] and seniors’ monitoring and assessment [8,9,10]. To ensure that older people can be active and maintain their independence and autonomy for longer, new health and social care strategies are being introduced. The World Health Organization (WHO) has proposed the Decade of Healthy Ageing, which began in 2021. Healthy ageing is defined by the WHO as ”the process of developing and maintaining the functional ability that enables well-being in older age” [11]. Healthy behaviours are key factors in active, healthy, and independent ageing as they delay or even reverse functional decline [4]. Finding an effective approach to active ageing is an international priority. One of the important aspects is to develop sustainable methods of cooperation between caregivers and the elderly and self-care to make seniors responsible and proactive in managing their own health. It is essential to empower seniors to take control and play an active role in monitoring and maintaining their health [12,13,14]. For example, they can use applications supporting medication administration, monitoring blood pressure, scheduling medical appointments, and promoting exercising and socializing [15,16,17,18,19].

Technology can alleviate the burden on traditional healthcare services in supporting healthy behaviours, for instance, in the prevention and treatment of chronic diseases [5,17,18,20,21], as well as the collection of data in daily life, to evaluate patterns and detect domains that require intervention [6,10,22,23,24,25]. Technologies can help with functional evaluation [9,10,18,25], promoting and motivating people towards health-related behaviours [6,17,18,20,21,26,27] and aiding in independent living [22,23,24,28,29,30,31]. The current state of the research literature demonstrates that there is a very diverse range of research on the use of technology and digital strategies to enable active and independent ageing [5,6,8,26,29,32,33,34]. Research varies in terms of applications and technologies. The most popular applications are communication, monitoring, and the promotion of health-related behaviours. Telecare and telemedicine have emerged with the spread of personal computers, the Internet, and mobile phones and include, e.g., remote consultation with patients, a remote collaboration between clinicians, and sending reminders, notifications and alerts [5]. Technological advances in recent decades have resulted in smaller and less intrusive monitoring devices that are suitable for use in daily life, as well as efficient algorithms that are able to deal with complex and large datasets. The technologies most commonly used to monitor and engage the elderly are computers [21] and mobile devices [9,20,22,28,35], wearable sensors [10,23,36,37,38], ambient sensors [18,23,24], virtual reality systems [15,17,25], and robots [8,39]. The monitoring of the elderly includes two areas: the monitoring of the body [10,15,21,37,40] and the monitoring of the mind [9,15,25]. There is also continuous monitoring [10,22,24,35] and monitoring during interventions (e.g., when using mobile applications) [9,15,21]. Monitoring is used in functional assessment for the early detection of diseases and health deterioration [9,10,15,18], supporting independent living [22,23,24,28,29,30,31], evaluating the performance of promoted behaviours [15,20,21,25], and creating awareness about a current behaviour, which can motivate healthier behaviours [18]. Furthermore, monitoring can provide new sources of information and insights to help researchers, caregivers, and policymakers implement better tactics to prevent a physical, cognitive and social decline in older adults [6]. Promoting, motivating, and supporting health-related behaviours is another common use of technology to adapt to the ageing population [15,17,18,20,21,26,27]. The combination of monitoring and assessment with digital applications has revolutionized interventions for the cognitive, physical, and social behaviours of older adults. Playing games and having fun while performing physical and cognitive activities is one of the research areas [15,17,20,21]. Another one is the applications that support seniors in ADL [18,26,28].

Despite the advantages of adopting new technologies, the elderly are slower to adopt them compared to younger adults [41]. In recent decades, the context of technology adoption by older adults has been frequently studied [42]. Some of the factors limiting access to new technology by the elderly such as the barriers of cognitive, physical, and, in some cases, financial origins have been recognized [16,28,38]. There are also some suggestions on how to deal with these obstacles [43,44]. An important aspect is the participation of the seniors in the process of identifying, conceptualizing, and designing digital services. These services then become better tailored, better identify key challenges, and are more relevant and meaningful [45]. Three approaches to user participation can be distinguished: User-Centered Design, Participatory Design, and User Innovation [46]. What they have in common is an understanding of the essence of the problem at hand, which influences the solutions adopted [47] and therefore can influence the design of the process of monitoring the psycho-physical condition of seniors.

Currently, we are seeing an increase in the acceptance of technology and a decrease in fears related to the ubiquity of technology (e.g., smartphones), including those aimed at seniors. Smartphone and smartwatch adoption by the general population has increased rapidly in recent years, including among the elderly population. More and more, people are carrying them routinely as they perform their daily activities. In Poland, according to the Centre for Public Opinion Research (CBOS) data,in 2015, smartphones were owned by 6 percent of people aged 65+ [48]; in 2017, 14 percent [49]; and in 2021, 36 percent [41]. We can expect a continuation of this upward trend, especially since in 2017, smartphones were used by 34 percent of the people aged 55–64, while in 2021, it was 72 percent [41,49]. It can be expected that with the increase in their application and spread, smartwatches will also become popular among the elderly. The global technology intelligence firm ABI Research expects continuous growth in the smartwatch market until 2027 [50].

Our project is part of the research on the promotion and motivation of the elderly towards pro-health behaviours and the monitoring of their activity to assess their condition. The aim of this work is to identify information that can be obtained, for the different recipients, on the behaviour of older people during cognitive, physical, and social outdoor activities, based on the proposed urban health paths, and how they can be used in further analysis. As a consequence of using a mobile application to guide users in the urban space and sensors for monitoring the heart rate and location of seniors, we were able to combine data from multiple sources to obtain meaningful information. The sensor data were given context, which is usually problematic with continuous ADL monitoring. In addition, the scope of the monitored data is substantial thanks to bringing activity outside and making many possible behaviours available to seniors.

## 2. Materials and Methods

### 2.1. Urban Health Paths Project

The presented work is part of a project that aims for the psycho-physical activation of the elderly, their rehabilitation, and space therapy. Its goal is to create a comprehensive tool for guiding seniors along the designed urban health paths employing urban architecture (e.g., buildings, walls, bridges, stairs, and benches) and their features as an impulse to perform physical and breathing exercises as well as an introduction to local history and architecture information [51].

The idea for the project originated from the Active Recovery Foundation from Wrocław, which works for the benefit of oncology patients and promotes a holistic approach to therapy. The project is carried out by supervisors and students of the Faculty of Automatic Control, Electronics and Computer Science, the Faculty of Architecture and the Faculty of Biomedical Engineering of the Silesian University of Technology, supported by external experts in the field of rehabilitation, space therapy, and universal design. It also receives consultation by seniors, who are members of the University of the Third Age and Centrum 3.0—The Gliwice Centre, for non-governmental organizations.

At present, three urban health paths for Gliwice city (Gliwice, Market Square Path; Gliwice, Center Path; and Gliwice, Downtown Path) were proposed. Along each path, special points, called waypoints, were designated, i.e., places for performing selected physical and breathing exercises. The path makes use of architectural objects (e.g., tenement houses, monuments) to determine the location of waypoints, as well as to show their history. The waypoints were selected due to the architectural elements present in them (for example, door portals and arcades on the ground floor of a tenement house) suitable for use in physical exercises [52]. Details of architectural objects (e.g., rustication of elevations, pilasters, or reliefs in the form of coats of arms) are also used to explain certain notions in the field of architecture. There are also specified resting points allowing seniors to listen to the sounds of the city and relax with breathing exercises.

Physical exercises take into account the age, fitness, abilities, and limitations of the elderly. Possible dysfunctions of the musculoskeletal system and safety for a broad group of participants were also taken into account when developing kinesiotherapy parameters. The aim of the prepared set of exercises was to stimulate various systems, especially respiratory and proprioception (e.g., balance and coordination). Due to the wide target group, as well as the diversified level of fitness, each exercise is graded in terms of the difficulty of performance. Training accents were placed on mixed cardiopulmonary training. Improving the efficiency of the respiratory system leads to better oxygenation, e.g., muscles and brain, which is crucial in the elderly. In turn, the improvement of blood flow in the blood vessels supports the work of the heart.

The proposed solution provides a model for using urban space in a conscious way while at the same time providing therapeutic and cognitive training. The developed solution not only improves general fitness but can also attract residents and visitors interested in the history and architecture of the chosen city.

### 2.2. Smartphone Application

To support the project’s goal, a smartphone application was created to guide and locate users in the urban space, as well as to provide additional architectural, historical, and health information and to present cognitive, breathing, and physical exercises. The application is dedicated to the elderly, so special attention was paid to its intuitiveness, resistance to user errors, and accessibility [53,54]. The application has only the necessary functionalities. The icons used in the application are understandable to an older person with clearly defined functionality. All issues related to gesture control or user input have been adapted to the needs of the elderly. The user interface is legible with adjustable parameters such as font size or colour scheme.

Every window in the application consists of three parts: the title bar, the part containing the contents of the window, and the navigation part with control buttons (Figure 1).

The user has access to all functionalities of the application from the main window (Figure 1a): they can change the application settings, set and view the user profile, receive help about the application and its use, and start exploring designed health paths. Two methods have been adopted in the application for the exploration of health paths. The first, the online mode, assumes that the senior moves around the city and interacts with the urban space. The actual location of the user (the latitude and longitude, as well as the direction in which the user is going) is monitored. The second method, trial mode, moves the user virtually from waypoint to waypoint on the selected path. Such a solution allows the user to get acquainted with the path at home, which means that they will feel more confident when going on the path for the first time. After selecting the exploration method, the senior can choose a path described by distance, the number of waypoints, and the route marked on the city map. After that, the application guides the user along the health path, switching between the path window and the waypoints windows. The path window (Figure 1b) contains a map with the marked path route, waypoints, and the user’s approximate location (for online mode). The map area can be moved and zoomed in and out. It is also possible to centre the map at the waypoint or the senior’s location. Additionally, at the beginning of the path, in online mode, the route to the first waypoint from the user’s location is presented. The Mapbox map services [55] are used to create and display maps and to locate the direction to the first waypoint from the user’s location. The control buttons in the navigation part of the path window allow seniors to go to the main window, receive help, end the health path, or go to the current waypoint window. For each waypoint, designers can prepare physical, breathing, and intellectual exercises (concerning the location’s history and architecture). The application allows the presentation of information in the form of images, audio, videos, and text. The relevant information is shown in the waypoint window (Figure 1c) when seniors start the exercise by pressing the exercise button for each exercise category. The end of the exercise is also confirmed by pressing the exercise button. From the waypoint window, the user can also end the waypoint exploration or cancel the waypoint exploration by returning to the path window. Confirmation is required to select the waypoint start, end, and path end. Upon completion of the path, a congratulatory window appears informing of the distance travelled and the waypoints completed. Seniors can also share this information with their contacts on WhatsApp.

To calculate the distance between the user and the waypoint, we use the equation for the distance between geographic locations (x—latitude, y—longitude) of 2 points A and B, ignoring the curvature of the Earth: (1)$$dist_{AB}=\sqrt{{\left(x_{B}-x_{A}\right)}^{2}+{\left(\cos\left(\frac{x_{A}\cdot \pi }{180}\right)\cdot \left(y_{B}-y_{A}\right)\right)}^{2}}\cdot \frac{40075.704}{360}$$

Information about paths and waypoints, as well as interface schemes, is stored in .json files that are parsed at the start of application execution. Each exercise, waypoint, and path has an ID that identifies it. Each media file used in an exercise, waypoint, or path description also has an ID. In the description of the exercise, we define, among other things, its category and subcategory. There are three main categories of exercises: motoric (physical), sensorial (breathing), and games (intellectual exercises). We can also define subcategories for each main category (for motoric exercises, they define which parts of the body the exercise is for, e.g., arms or legs). We can specify several levels of difficulty for each exercise. For each level, its description and difficulty are defined. Some exercises can be performed with varying degrees of difficulty, so we use a difficulty range instead of a single value. We use two values: the minimum and maximum difficulty of the exercise. For example, in one of the exercises, the senior has to lean back, imitating the arch of an arcade; the difficulty of the exercise changes with the depth of the arch of the back. If the exercise has a precise difficulty, the maximum and minimum values are the same. We have decided that the exercises can fall into the difficulty range of 1–5. Each waypoint description contains a list of exercise identifiers. Each path description contains a list of waypoint identifiers.

The application is dedicated for Android mobile devices and was developed in the graphical environment Unity using Mapbox SDK, Unity Cross Platform Native Plugins Essential Kit (for WhatsApp integration) and Newtonsoft JSON (for .json file management).

### 2.3. Sensory Data

The developed mobile application itself does not allow for a detailed analysis of the impact of the health path on the physical condition of the elderly and thus makes it challenging to identify potential problems with the path; for example, exercise can be too strenuous for the elderly. To obtain the relevant data, we decided to test a wearable device—a smartwatch for monitoring the selected sensory data. In particular, information on the senior’s heart rate can significantly expand our knowledge of the senior’s physical condition during the path exploration. The heart rate is the number of times that a heart beats in a minute. A healthy heart speeds up and slows down to accommodate a person’s changing need for oxygen as their activities vary throughout the day. The heartbeat becomes faster when people are active, excited, or scared and drops when they are resting, calm, or comfortable. Furthermore, heart rate depends on a person’s age and overall health. Therefore, a “normal” heart rate varies from person to person, but for most healthy adult women and men, resting heart rates (when they are are at rest and their heart is pumping the lowest amount of blood to supply the oxygen your body needs) range from 60 to 100 BPM (beats per minute). The rate at which the heart beats when it is working its hardest is a maximum heart rate and can be roughly calculated from a person’s age (that is, 220 – age). Smartwatches and fitness bands measure heart rate by scanning blood flow near a wrist with green LED lights paired with light-sensitive photodiodes that illuminate the skin and measure changes in light absorption (when the heart beats, the bloodflow in the wrist—and the green light absorption—is greater than between the beats). Smartwatches are not as accurate as professional medical equipment, but they can be pretty close and let an individual monitor their current heart rate while they are walking, running, cycling, or exercising.

Since managing two applications at the same time can be cumbersome and discourage seniors from using the smartwatch for health paths, automatic switching on and off of the smartwatch application is applied, as well as managing the measurement of the desired parameters without user intervention. Work on the implementation of the sensory data measurement system began with the creation of a wearable application for the tested watch, Samsung Galaxy Watch 4, acquiring data from built-in sensors. The data are manually extracted by creating a watch exercise, related to the health path, monitoring the following parameters: heart rate, calories, distance, and the number of steps. The result of the work is a program written in Kotlin for Android Wear OS for which the following solutions have been implemented:Automatic switching on of the application on the wearable device when the user starts the health path in online mode and automatic switching off of the application at the end of the path.Manual launch of the application on the wearable device whenever the user wants.Low-level communication system with a paired phone using the Bluetooth standard for the correct identification of the data recipient and data transmission. Data are sent every 3 s.The system can be paused by pressing a button on the screen; the application can be paused and then restarted in the event of, for example, a break during exercise.Energy-saving system: the screen works in ambient mode for the duration of the exercise, preventing the interface from refreshing and turning off the screen when the user does not raise their hand. This solution allows for a significant extension of the working time of the created application on a single charging of the smartwatch.The ability to run the application in the background allows the user to use the rest of the applications on the smartwatch without interrupting the exercise.

In addition, for our smartphone application, a Java plugin was created, which receives data from the smartwatch via Bluetooth communication and forwards it to the application for further processing.

The exercise on the smartwatch starts when the senior chooses to begin the path exploration in online mode in the mobile application or when they manually turn on the exercise on the smartwatch. The smartwatch exercise stops when the senior completes the path in the mobile application or manually turns off the exercise. During the exercise, every 3 s, the smartwatch saves information about the last heart rate measured by the sensor and accumulated since the beginning of the exercise, which calculated on the smartwatch: calories burned, distance travelled, and steps taken. These data are then sent to our smartphone application. At the moment, the collected data are transferred to the database, and the aggregated information is displayed in the application’s congratulatory window.

### 2.4. Data Storage System

In the project, we decided to save the data generated by the mobile application and the smartwatch in the external database. The recorded data have been divided into the following categories:User—user and mobile device information (user profile, current application settings).Settings—information on how the mobile application is operated by the user (e.g., information on adjusting the application settings, such as font size or colour scheme, to the user’s needs).User actions—information on the user actions performed in the mobile application during the exploration of health paths (e.g., button pressed, path selected ).Health data—parameters monitored by the smartwatch (heart rate, calories, distance, number of steps).

Users are identified on the basis of a unique device identifier. To ensure anonymity for seniors and the ability to identify new information from the same user, we decided on the hashing device ID with the SHA256 hashing function with a salt. For this reason, the user is recognized by the database on reconnection, but the database does not store information that would allow the identification of the user’s device, maintaining the user’s anonymity.

An entry to the database of the user’s actions in the mobile application contains the following information: device ID, action name (usually “Button Clicked”), action description (e.g., waypoint started/completed), path ID, waypoint ID, exercise ID, date and time of the entry (in the format “yyyy-mm-ddThh:mm:ss”), and device position ( latitude and longitude). The following types of senior activities are recorded:Paths-related.Consecutive steps to start the health path are recorded: “go to path list”, “show path details”, “start new demo path”, and “start new path”. If the user returns to the main window while executing the path (“return to the main window”), they can choose between two actions: “cancel path” or “continue path”. Users can select “cancel path” at any time during path exploration. If the user reaches the end of the path, an action “path completed” is recorded. At the end of a completed or cancelled path, information about the finished path is saved (e.g., the number of points scored, and distance travelled).Waypoints-related.There are four actions that the user can perform regarding waypoints that are not related to waypoint exercises: “set waypoint”, “cancel waypoint”, “complete waypoint”, and ”listen to the fun fact audio about waypoint“.Exercises-related.Information on the beginning and end of an exercise is recorded for each of the three main categories: ”start/complete exercise, category: Game/Motorical/Sensorial ”.

An entry to the database with the parameters sent by the smartwatch contains the following information: device ID, date and time of the database entry recorded with precision in seconds (in the format “yyyy-mm-ddThh:mm:ss”), Unix epoch time generated by smartwatch with precision of milliseconds, heart rate in BPM, burned calories in kcal, distance in meters, and the number of steps. The last three values are calculated from the start of the health path or the launch of the smartwatch application.

For this project, it was decided that the recording of data in the external database must be completed automatically so that the senior does not feel discomfort while saving data. Additionally, the data should be sent when the mobile phone has a stable internet connection, which is not always possible when the user is outdoors. The data-recording process takes place in two steps. Initially, the mobile application saves all the data in the local Realm database. Then, after obtaining a stable internet connection, it automatically synchronizes its data with the MongoDB Atlas multi-cloud database with the use of Device Realm Sync. The Device Realm Sync technology developed by MongoDb in cooperation with the company responsible for the Realm mobile database enables the automated synchronization of documents in database collections on a mobile device with a cloud database only when the mobile device obtains a stable internet connection. This solution ensures that the experience of users on mobile devices is responsive and performant, regardless of their current network status and the mobile application continues to function even when disconnected from the network [56].

## 3. Results

As we showed in Section 2, a lot of data are generated by the mobile application and wearable device, i.e., the smartwatch. Combined, they can provide information on seniors’ psycho-physical condition that can be used in many applications. We distinguished three types of data recipients: seniors, path/exercise designers, and the public. By the last group of people, we mean representatives of social and state organizations and health services, whose decisions may affect the seniors’ quality of life. This project is at an early stage. The data we have acquired are incomplete and unrepresentative. The seniors who tested the tools constitute a small group that is physically and intellectually able and ready to face new challenges. It is difficult to say how they look compared to the rest of the seniors who can be persuaded to navigate the health paths. This is something we are going to investigate. In this work, we identified information that can be obtained for the different recipients, and how it can be used in further analysis.

### 3.1. Exercises

For all exercises, we have information on how long the exercise lasted and what the heart rate was. We calculated four values for the heart rate—average, maximum, minimum and standard deviation—which allow us to evaluate the exercise performance. To increase the precision of the assessment, we can determine the average exercise heart rate relative to that recorded for the individual average heart rate during walking or physical and breathing exercises. For example, Figure 2 shows the heart rate graph of a person while resting, walking, and exercising. The average heart rate for walking was 102 BPM; for breathing exercises, it was 80 BPM; and for physical exercises, it was 139 BPM. For the evaluated exercise, the average heart rate was 120 BMP. We can use recorded values for the person and determine that it is 118% of the average heart rate for walking, 150% for resting, and 86% for exercising. The percentages allow for the comparison of heart rates for people of different ages and physical conditions.

In addition, exercises are marked as passed or failed. The exercise is considered passed if its duration is greater than the minimum exercise time threshold *eMinTime* (set to 1 min). This is the minimum time a senior should spend on each exercise. This rule allows discarding exercises only scrolled in the application without real commitment.

Exercises can be divided depending on the category, subcategory, and range of difficulty. Additionally, the performed exercises can be differentiated depending on the profile of the exerciser. For instance, we can calculate statistics for leg exercises with a difficulty level of 1–2 for women over the age of 70. We can compare the calculated statistics with each other. Moreover, we may find that, on average, arm exercises have a higher heart rate (intensity) than leg exercises, regardless of the exerciser’s profile. We can also calculate the correlation for the characteristics of exercises that have successive levels (e.g., level of difficulty, age of the exerciser); for example, the older people are, the shorter the duration of all exercises.

Furthermore, the path designer evaluating a particular exercise may be interested in comparing the statistics for the given exercise (with specific exercise ID) with the statistics of similar exercises. Information showing that users perform a given exercise at a lower intensity than exercises of similar difficulty and the same subcategory or that many users skip the exercise altogether may indicate that the exercise needs refinement. On the basis of the results obtained, the designer of the exercise may also conclude that its degree of difficulty has been overstated or underestimated. When designing exercises, more general results can also be used, such as comparing exercises with different characteristics. If seniors show a preference for exercises with specific characteristics, the number of these exercises can be increased or other exercises can be modified to include elements that may encourage seniors to exercise (this applies to all types of exercise).

Moreover, a senior may like to know how their performance in a particular exercise or sets of exercises compares to their previous results. A senior may receive, for example, the following information: “You exercised 2 min longer than your average”.

General statistical information about exercises (comparisons, correlations) can be used by the public in many applications, for example, to determine the need for types of equipment in outdoor gyms. Knowing the profile of seniors ready for new activities, other initiatives can be offered to them, but the public can also look for activation methods suitable for other groups.

Information about the psychophysical condition of seniors can be assessed not only on the basis of physical exercises. The observed decrease in heart rate during breathing exercises allows the participants’ ability to relax to be assessed. The time spent on the City Game and the results of architectural and historical tests allow the intellectual state of the elderly to be assessed.

### 3.2. Paths

Seniors can explore the paths in trial mode or in an online one. Most of the analyzed data are generated in online mode. For the trial mode, we used the viewing time, which we add to parameters of specific paths (with the same pathID) carried out up to a month after the trial version. The information showing that the path is being executed again is also set as a path parameter. The number of path executions up to one month before the execution of the current path is set.

The path is based on waypoints and exercises, so we use their statistics to define the path. For exercises, we use the exercise statistics described in Section 3.1. As a standard, we count the passed physical exercises, breathing exercises, and City Game exercises and save their number and percentage as the path parameters. In the case of additional needs, the researcher can count exercises of a given type, for example, passed physical exercises for legs performed by people aged 70–75. Their average execution time can also be utilized as well as other values (e.g., minimum heart rate). In the case of waypoints, we are primarily interested in information on whether the senior was in the area of the waypoint (whether the waypoint can be considered passed). Setting a waypoint as passed consists of calculating the distance between the senior’s location and the waypoint using Equation (Equation 1) and then checking if it is less than the set threshold distance *wMaxDist*. Deciding on the *wMaxDist* value is not easy, as there are several factors that determine the location of the senior we receive. First, there is an issue with the accuracy of the phone location system in various environments. Users use different phones (with different methods of estimating their location and different signal strength from the Global Positioning System (GPS) and the Internet), which estimate their location with differing levels of accuracy [57].

Second, for many waypoints, the right position of the senior is not obvious: it is more important whether the senior has eye contact with the object according to which they are exercising. This may mean that multiple locations will be valid for a given waypoint. Figure 3 shows an example situation at a Rzeźniczy Square waypoint on the Gliwice Downtown Path, where the entire square is used for exercises. The target of the waypoint was set on one side of the square, and the senior indicated that they are at the waypoint, when they were in fact on the other side. In this tiny square, the difference in distance was 26.48 m. In addition, if seniors exercise in a group, they may be more spread apart so as not to disturb each other. In view of the above issues,we decided to set the *wMaxDist* value to 50 m. We decided on a higher value, also taking into account the nature of the application and its recipients. If seniors are in the vicinity of the waypoint and indicate that they have reached it, their perceptions should also be considered. Additionally, we determine whether the waypoint has been explored, i.e., whether at least one exercise has been set as passed within it. For each path, we set the number and percentage of waypoints that were passed and explored. A path is considered passed (taken into account in the comparisons and calculations) if the percentage of passed waypoints is greater than 0% (at least one waypoint has been passed).

For each executed path, its time, distance travelled, walking time, and the number of steps are also calculated. We also want to know how much time seniors rest during path execution. The aggregated value of rest time is of interest, as well as the average, minimum, and maximum values for each rest break and their numbers. For example, for a 1.5 h path, 40 min can be for rest, with six breaks that are on average 7 min, with the longest lasting 15 min and the shortest 3 min.

With these data, we can generate path statistics that can be used by all proposed data recipients. For example, path designers can use information about rest breaks to design paths that offer rest areas at chosen intervals. They can also adjust the number of waypoints and the distance of the path to seniors’ preferences. It may be interesting to know how many participants go only for virtual walks and what their profile is. Interviews with people with similar profiles may suggest why they do not actively participate in exploring health paths. Seniors can use path statistics to compare their performance with other seniors, but also with themselves (e.g., to look for improvement). The public, from all these data, can extract information about seniors, e.g., their physical condition, but also, looking at the number of different paths executed by seniors, how willing they are to try new things (assuming more paths are available in the area).

### 3.3. Time and Location

By analyzing the online paths explored by seniors, we can also look at the time when seniors choose to start the paths and the area where the paths are located.

Information about the preferred season/months can be taken into account by the path designer when selecting waypoints and exercises. Some waypoints and exercises may not be appropriate when it is cold and seniors have a lot of clothes on, and others may not be appropriate when the temperature is high and the sun is hot. Insufficient shade at waypoints and on walks between waypoints in summer months can be too tiring and even dangerous for seniors. This may translate into path statistics, for example, a decrease in seniors’ attendance on given paths at certain times. Designers can analyze whether certain paths/exercises are more/less attractive at a given time and use this knowledge to devise new paths and improve the existing ones.

The days of the week and hours of the day spent by seniors on walks also carry information about the preferred time for outdoor activities and can be used by the public to organize a variety of activities aimed at seniors. To encourage seniors to leave the house, additional attractions can be prepared at selected times, as well as joint outings on dates preferred by seniors.

Another important piece of information is how often seniors go out on the paths: several times a week, once a week, several times a month, or once a month. With such information, we can investigate the reason for the observed frequencies. Does the number of paths available in the area matter? Are seniors willing (or even prefer) to walk the same path more times, or is there a need to provide more variety of paths? We can also look at changes in recorded patterns. Seniors who actively used health paths stop, and others who only viewed the virtual paths go out on the paths. We could explore what motivates them and what has changed. By observing the generated statistics, researchers can detect situations involving seniors that are relevant to the public and worth further investigation.

The health paths are prepared for specific areas and dedicated to local residents. This allows the comparison of people from different areas based on the generated statistics. One can imagine that the results recorded for communities from different areas and even countries would allow one to compare their physical condition, attitude to outdoor activities, and the use of mobile technologies.

### 3.4. Social Data

Interviews with seniors, members of the University of the Third Age, and Centrum 3.0 showed that they feel better when performing tasks in groups. They feel safer and are less embarrassed and self-conscious when exercising in public.

As part of data analysis, we can identify the paths explored in groups and compare them to the paths executed alone. On the basis of data, such as time and location, recorded for subsequent waypoints by different participants, we can identify the paths carried out in pairs or groups. To find paths performed by groups, we first search for paths in online mode with the same path ID and start date. Then, for each waypoint in the path, we check the locations registered by the seniors. If the distance between them $diffLocation_{AB}$ (calculated with Equation (Equation 1)) is less than sMaxDist = 30 m, a social distance factor sDist is set to 1; otherwise, each meter above sMaxDist reduces sDist by sMeter = 0.05, down to 0 (Equation (Equation 2)). Next, we look at the time intervals for each waypoint and if the time intervals collide, the social time factor sTime is set to 1; otherwise, each additional minute between the time intervals $diffMinutes_{AB}$ decreases the value of sTime by sMinute = 0.1 until it reaches 0 (Equotation (Equation 3)). For example, for two time intervals, <13:40:20,13:55:13> and <13:35:40,13:42:27>, we have a collision, so we set sTime to 1, and for intervals <13:40 :20,13:55:13> and <13:35:40,13:37:45>, the difference is more than 2 min, so sTime is equal to 0.8. Finally, we calculate the acceptance value $sA_{AB}^{w}$ of seniors *A* and *B* meeting at the waypoint *w* using the equations
(2)$$sDist=max(1-sMeter*max\left(diffLocation_{AB}-sMaxDist,0\right),0)$$
(3)$$sTime=max(1-sMinute*diffMinutes_{AB},0)$$
(4)$$sA_{AB}^{w}=sTime*sDist$$

The social path acceptance value $sA_{AB}$ of the seniors *A* and *B* is calculated based on the $sA_{AB}^{w}$ values obtained for all waypoints of the path using the equation: (5)$$sA_{AB}=\frac{\sum_{w=1}^{W}sA_{AB}^{w}}{W}$$
where *W* is the number of waypoints on the path.

If the $sA_{AB}$ value for a given path and seniors *A* and *B* is greater than the social acceptance path threshold sAt, the path is considered to have been made jointly by seniors *A* and *B*. For now, the sAt has been set to 0.6. The sMaxDist, sMeter, sMinute, and sAt values have been selected based on the authors’ knowledge and verified during tests with seniors, members of the University of the Third Age and Centrum 3.0.

The next step is to specify a group of paths with a specified path ID and start date. For a path to be added to a group, its $sA_{AB}$ value must be greater than sAt with at least one path in the group. Finally, for each path, we set a parameter specifying how many seniors together explored the path (how many paths are in its group minus one). If the path does not belong to any group, the value is set to 0.

Then, we can compare the statistics for the paths performed in groups or alone. We can check whether there are differences between groups of paths, e.g., whether there are weaker and stronger groups. We can also evaluate the profiles of seniors exercising in groups and alone, as well as those who form individual groups. It can be seen if people with similar profiles walk and exercise together. Furthermore, it can be checked whether seniors belong primarily to one or the other group: those who make paths alone or in groups (e.g., by calculating the rate of trips alone by dividing the number of paths completed alone by all paths for a given senior).

Health path designers can use this information, for example, to identify paths/exercises that can be completed alone to extract data for connecting seniors who are not part of groups with those whom they can exercise with. The information showing that a certain number of lonely seniors follow the same paths can be used by public organizations to organize joint outings and increase the socialization of seniors.

The paths whose social path acceptance value $sA_{AB}$ is below the acceptance path threshold sAt, but above a certain value, e.g., 0.4–0.6, may be of interest for analysis. For example, some seniors started the path together and some gave up. Why? What was the profile of those who gave up and what was the profile of those who persevered? The answers to these questions can help designers when designing paths/exercises but also the public in predicting the behaviour of seniors in various social situations.

### 3.5. Sensory Data

The presented project relies heavily on user input, so we assume that mistakes will be made. For example, a senior may forget to select information about reaching a waypoint or starting an exercise. One of the important goals of monitoring the behaviour of seniors in our project is to detect behavioural patterns and determine the psycho-physical state of seniors on their basis. Information about mistakes (for example, the number of failed waypoints caused by selecting them later in the path exploration) is equally important and worth analyzing as well-executed paths.

Data generated by sensors, such as heart rate or registered locations, can additionally help in assessing the behaviour of seniors and verify the data they enter. As part of the initial tests conducted with seniors (members of Centrum 3.0) along the Gliwice Downtown Path, we checked their usefulness in our project.

First, we checked whether some of the data sent between the smartwatch and the smartphone were lost, i.e., whether the phone is able to download data from the smartwatch on an ongoing basis (every 3 s). Figure 4 shows the differences between times of the smartwatch’s successive sensory data (logged with milliseconds precision) stored in the project database, which were recorded during approximately 18 min of exercise. As we can see, all values are close to the required 3 s. Other studied values are times recorded by the mobile application (with precision down to a second). For the first 10 min, the results (Figure 5) correspond to the results in Figure 4 (taking into account the lower precision of the mobile application); then, there is a 7 min difference between the subsequent recorded data, and the remaining 159 data are logged at the same time. To save the battery, a 10 min interval of user inactivity in the application puts it to sleep. The next registration took place after the application was woken up (when the user entered information about the end of the exercise). The discrepancy between the time of the sensory data and the time generated in the application requires the use of the former when determining the time for the smartwatch data. The requirement to synchronize the smartphone and smartwatch via Bluetooth ensures that the time of the smartphone and smartwatch is synchronized. In the test, the start time and the end time of the exercise on the smartwatch and the smartphone were the same.

We also checked the usefulness of using the heart rate to determine the intensity of the senior’s motion as well as their state of relaxation. We were interested in whether the recorded heart rate values for different behaviours (resting, walking, intense exercising) for the same person would differ enough to be useful for the evaluation of seniors’ behaviours. Figure 2 shows the heart rate of the tested person. The average heart rate for walking is 102 BPM, for breathing exercises it is 80 BPM, and for physical exercises it is 139 BPM. The differences are large enough to classify the behaviours of seniors.

The location of the senior on the path is used often in our project. Many results depend on the accuracy of this value. During tests conducted by five seniors along the Gliwice Downtown Path, the average distance from seniors (during pressing the start of the waypoint button) to waypoints was 22.34 m (with a standard deviation of 15.5 m). Importantly, none of the values obtained during tests exceeded the set threshold distance wMaxDist (50 m). In addition, all paths were included in a single social group of paths.

Preliminary studies confirmed the usefulness of the tested sensory data. The number of people testing and the amount of data obtained were too small to draw categorical conclusions, but the current results are encouraging.

## 4. Discussion and Conclusions

In this paper, we have presented a project on the motivation of seniors to perform physical and cognitive exercises while exploring a urban space based on the proposed health paths. The behaviour of seniors is monitored during their activities performed using wearable sensors (location, steps taken, distance covered, and a heart rate) and a mobile application (information on initiated and completed actions and achieved results) that guides seniors on their health paths. The proposed project is part of the research on active ageing with a focus on monitoring seniors’ activities and motivating them to undertake pro-health behaviour. The result of the implemented project is the development of a multidimensional method of collecting data as part of an innovative approach to encouraging activity in seniors; this includes having seniors exercise outdoors while using the available urban space with its history and architecture, to mobilize seniors to perform physical and cognitive tasks, allowing them to be flexible and autonomous while maintaining a rigid framework implemented in the application. Seniors can adapt their behaviour to their limitations and needs within predetermined scenarios. This definition of acceptable seniors’ activities translates into a wide range of diverse data from which short- and long-term patterns can be drawn. We have shown that by combining sensory data and mobile application data, we can provide context to the sensory data, thanks to which we can formulate more insightful assessments of seniors’ behaviour. Preliminary studies have confirmed the usefulness of the sensory data adapted in our project. In this paper, we also distinguished three types of data recipients (seniors, path/exercise designers, and the public) for whom we have presented the methods of analyzing the data received and examples of their use.

There is a vast area for further research. One of the directions of research concerns increasing the amount of data collected, which will allow for a better assessment of seniors’ behaviour. Further work on the use of sensory data will allow a more detailed evaluation of exercise performance by seniors. We could also introduce surveys to mobile applications to profile users in more detail (e.g., health problems, number of close friends). Another direction of research may concern the social problems of seniors. We could add to the mobile application the possibility of creating groups of seniors who can arrange trips together, exchange messages, or engage in friendly competition.

## Figures and Tables

**Figure 1 sensors-23-01602-f001:**
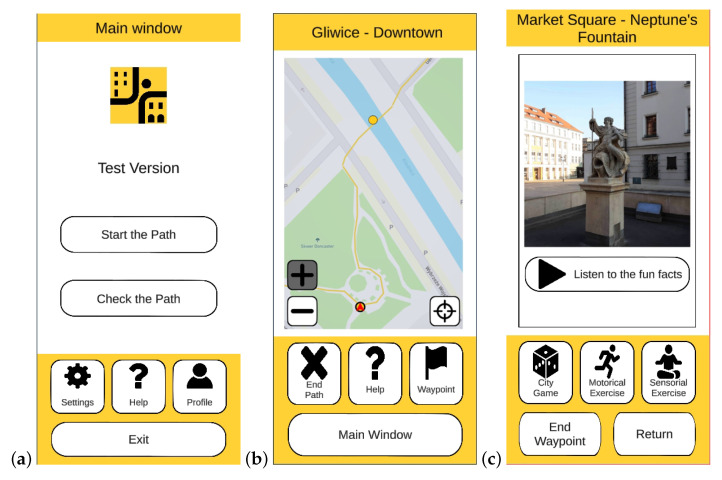
Screenshots from the mobile application *Urban Health Path:* (**a**) Main window, (**b**) Path window (an example), (**c**) Waypoint window (an example).

**Figure 2 sensors-23-01602-f002:**
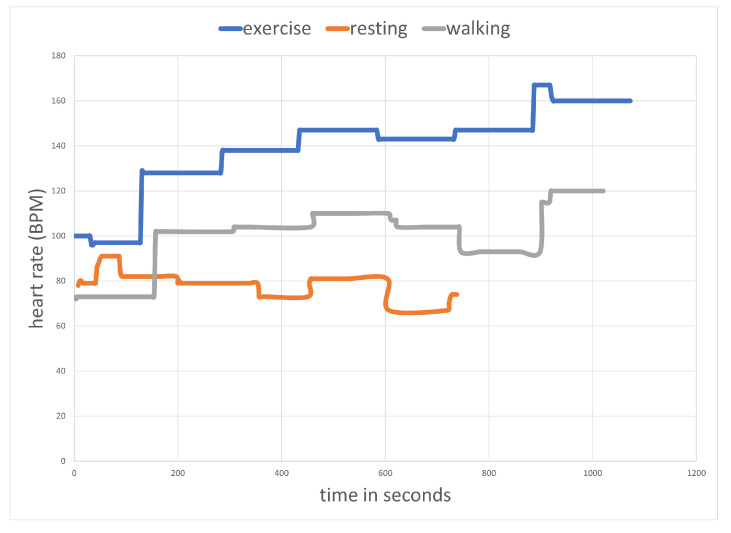
Example of heart rate statistics.

**Figure 3 sensors-23-01602-f003:**
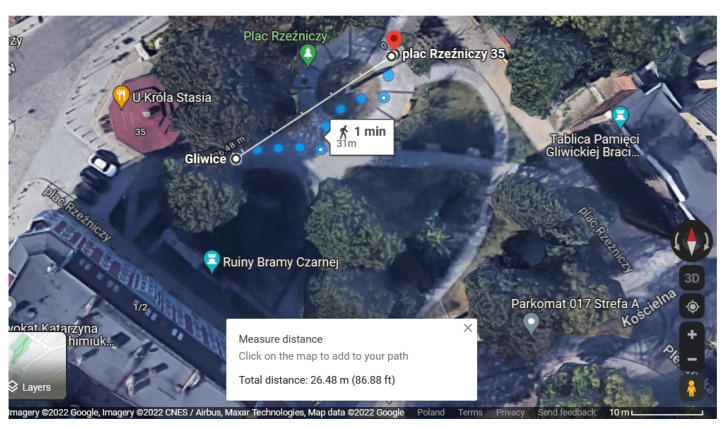
The distance between the senior’s location and the waypoint for the Skwer Rzezniczy waypoint on the Gliwice Downtown Path (source: Google Maps/Google Earth, 2022).

**Figure 4 sensors-23-01602-f004:**
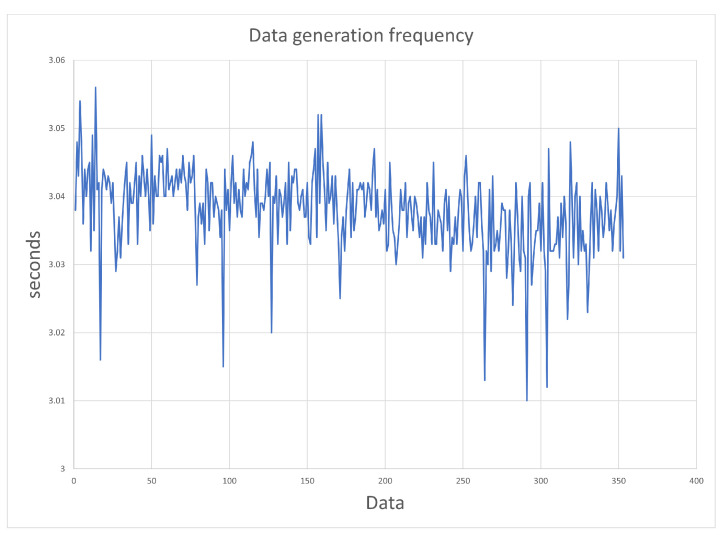
The frequency of generation of sensory data of the smartwatch (Samsung Galaxy Watch 4) received by smartphone (Samsung Galaxy S20+ with Android 12).

**Figure 5 sensors-23-01602-f005:**
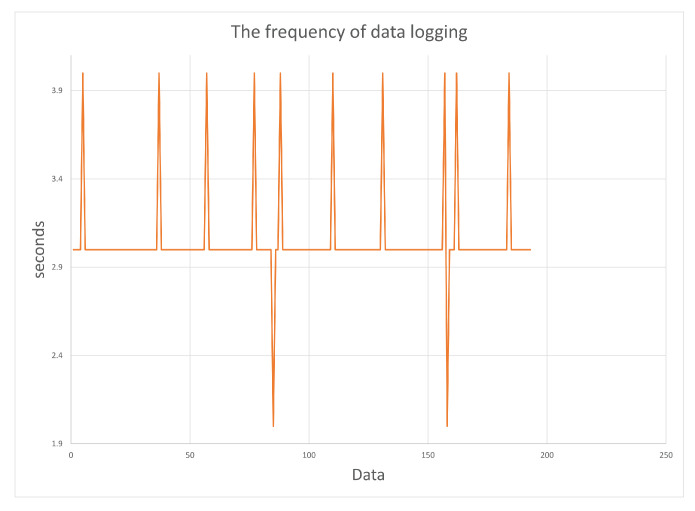
The frequency of data recording in the mobile application Urban Health Path.

## Data Availability

Not applicable.

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
