# Peer review of "The Problem of Monitoring the Psycho-Physical Condition of Seniors during Proposed Activities in Urban Space"

_sensors, 2023, doi:10.3390/s23031602_

Round 1

Reviewer 1 Report

This paper studied the problem of monitoring and promoting the health-related behaviour of seniors. They introduced a comprehensive tool for guiding seniors along the designed urban health paths employing urban architecture as an impulse to perform physical and cognitive exercises and used wearable sensors and mobile application which monitoring the behaviour of seniors during activities. However, there are some issues that should be addressed.

Major comments:

1. In part ABSTRACT, the introduction to technology should be supplemented, so that readers can better understand the authors’ intention.

2. In part 2.3, authors used a smartwatch to get the health-related data, such as heart rate, calories, distance, and the number of steps. Authors should supplement the calculation process and then discuss the result of the calculation. In this case, I think the manuscript would fit the SENSORS scope more relevantly.

3. The logical structure of the manuscript is incomplete, there is a lack of part “Conclusions”.

4. The current manuscript needs to be polished by a native English speaker or a professional language editing service.

Minor comments:

1.      Some sentences contain grammatical mistakes or are not complete, such as, in line 54, this sentence lacks a predicate “are”. Please check the manuscript carefully.

Reviewer 2 Report

The topic presented is certainly important: the fight against both physical and psychological aging is a matter that will increasingly affect the most developed societies.

The authors show that they have mastered the scientific literature on the topic under examination (45-46). At the same time the project in question is described several times with great accuracy. However, the passage "from curative to preventive care" (34) could be further developed precisely by virtue of its relevance for the transformations that the welfare state will undergo. Similarly, the reference to the individual as the protagonist of his own health (34-35) can be further developed: notes could therefore be added that recall the reference bibliography (which in this regard could be slightly supplemented).

The project presented is solid and has the advantage of being applied to a reality that already exists (that is, the city with its infrastructures).

To cite other of its strengths, the article has, on the one hand, the awareness of its intrinsic limits, and this can be seen both in the awareness of the literature on aging and on the strategies to contrast it (45-47). On the other hand, in the conclusion (553-560), the authors show that they are aware of the breadth of future research on the same topic. Furthermore, concrete ways are identified precisely with regard to this future research (556-560).

A point that could be further clarified concerns the objectives of the project: it intends to monitor both the body and the mind (58). Well, it seems to me that the mind is monitored more for the benefits towards the body than for the advantages for the mind itself: this is a point that could be clarified further. Instead, I would avoid mentioning the (direct and indirect) competition between seniors (377-380), because it can be misleading. Competition, even indirect, between seniors lends itself to the objections of the possible stress that can derive from it and to the implicit desire to emulate other results.
